# Structure and inference in hypergraphs with node attributes

Anna Badalyan[1], Nicolò Ruggeri [1,2] ✉ & Caterina De Bacco [1] ✉

Many networked datasets with units interacting in groups of two or more, encoded with hypergraphs, are accompanied by extra information about nodes, such as the role of an individual in a workplace. Here we show how these node attributes can be used to improve our understanding of the structure resulting from higher-order interactions. We consider the problem of community detection in hypergraphs and develop a principled model that combines higher-order interactions and node attributes to better represent the observed interactions and to detect communities more accurately than using either of these types of information alone. The method learns automatically from the input data the extent to which structure and attributes contribute to explain the data, down weighing or discarding attributes if not informative. Our algorithmic implementation is efficient and scales to large hypergraphs and interactions of large numbers of units. We apply our method to a variety of systems, showing strong performance in hyperedge prediction tasks and in selecting community divisions that correlate with attributes when these are informative, but discarding them otherwise. Our approach illustrates the advantage of using informative node attributes when available with higher-order data.

Over recent years, systems where units interact in groups of two or more have been increasingly investigated. Such higher-order interactions have been observed in a wide variety of systems, including cellular networks[1], drug recombination[2], ecological communities[3] and functional mapping of the human brain[4].

These systems can be better described by hypergraphs, where hyperedges encode interactions among an arbitrary number of units[5,6]. Often, research in this area solely considers the topology of hypergraphs, that is, a set of nodes and their higher-order interactions. Many hypergraph datasets, however, include attributes that describe properties of nodes, such as the age of an individual, their job title in the context of workplace interactions, or the political affiliation of a voter. In this work, we consider how to extend the analysis of hypergraphs to incorporate this extra information.

We focus on the relevant task of community detection, where the goal is to cluster nodes in a hypergraph. Community detection algorithms solely based on interactions tend to cluster nodes based on notions of affinity between communities, cluster separation, or other arguments similar to those classically utilized on graphs[7]. However, one can assume that relevant information about the communities and the hyperedge formation mechanism is additionally contained in the attributes accompanying a dataset.

For instance, students in a school have been observed to interact more likely in groups that involve individuals in the same classes[8]. A similar observation was also made for dyadic networks, where incorporating node attributes helped in community detection and other related inference tasks, e.g., prediction of missing information[9–13].

Several tools have been developed for community detection in higher-order data[14–17]. Methods based on statistical inference have established themselves as effective tools in this direction, as they are both mathematically principled and have a high computational efficiency[18–20].

Here, we build on these approaches to incorporate node attributes into a community detection framework for higher-order

[1]Max Planck Institute for Intelligent Systems, Cyber Valley, Tübingen, Germany. [2]Department of Computer Science, ETH, Zürich, Switzerland.
✉e-mail: nicolo.ruggeri@tuebingen.mpg.de; caterina.debacco@tuebingen.mpg.de

interactions. More precisely, we follow the principles behind generative models for networks, which incorporate community structure by means of latent variables that are inferred directly from the observed interactions[21–23] and extend them to incorporate extra information on nodes.

The model we propose has several desirable features. It is flexible, as it can be applied to both weighted and unweighted hypergraphs, it can incorporate different node attributes, categorical or binary, and it outputs overlapping communities, where nodes can belong to multiple groups simultaneously. Furthermore, the model does not assume any a priori correlation structure between the attributes and the communities. Rather, it infers such a connection directly from the data. The extent of this contribution can vary based on the dataset. In the favorable case where attributes are correlated well with the communities, our model exploits such additional information to improve community detection. This is particularly beneficial in situations where data is sparse or when data availability is limited to an incomplete set of observations. In less favorable situations where correlation is low (for instance when the attributes do not align with the mechanism generating higher-order interactions), the model can nevertheless either discard or downweigh this information.

In some cases, a system can be explained well by different community divisions. Our model allows selecting a particular community structure guided by the desired attribute, provided that it is informative, as measured automatically by fitting the data. This allows a practitioner to focus the analysis of group interactions on some particular node characteristic.

Finally, our model is computationally efficient, as it scales to large hypergraphs and large hyperedge sizes. This feature is particularly relevant in the presence of higher-order interaction, where the increased computational complexity limits the range of models that can be practically implemented into viable algorithms.

Few works are available that investigate community detection in hypergraphs in presence of node attributes[24–26], but they are limited to clustering nodes without providing additional probabilistic estimates. Furthermore, they can be computationally burdening, or they typically rely on stronger assumptions about the nature of the data (e.g., assume real-valued weights) or the communities (e.g., nodes can only belong to one group).

## Results

### The model

We propose a probabilistic model that incorporates both the structure of a hypergraph, i.e., the interactions observed in the data, and additional attributes (or covariates) on the nodes. These two types of information, which we call structural and attribute information, have been previously shown to be informative in modeling community structure in networks, when there is correlation to be exploited[9–12].

We denote a hypergraph as $H = (V, E, A)$, where $V = \{1, ..., N\}$ is a set of nodes, $E$ is a set of observed hyperedges whose elements $e \in E$ are arbitrary sets of two or more nodes in $V$, and $A$ is a vector containing the weights of edges. In this work, we assume that weights are positive and integer quantities. Denoting $\Omega$ as the set of all possible hyperedges, we have that $A_e$ is the weight of edge $e$ when $e \in E$, otherwise $A_e = 0$ if $e \in \Omega \backslash E$. Given these definitions, the observed edge set $E$ can equivalently be represented as $E = \{e \in \Omega \mid A_e > 0\}$. We represent the covariates on nodes as a matrix $X \in \mathbb{R}^{N \times Z}$, where $Z$ is the number of attributes, with entries equal to 1 if the node $i$ has attribute $z$ and 0 otherwise. We note that a node can have several types of covariates, e.g., gender and age, which are then one-hot encoded as attributes.

We model the presence of structural information $A$ and covariate information $X$ probabilistically, assuming a joint probability of these two types of information that is mediated by a set of latent variables $\theta = \{w, \beta, u\}$. Here $w$, $\beta$ are specific to each of the two distinct types of information, while the quantity $u$ is a latent variable shared between

the two. The presence of a shared $u$ is a key to allow coupling the two types of information and extracting valuable insights about the system. Formally, we assume

$$P(A, X \mid \theta) = P_A(A \mid w, u) P_X(X \mid \beta, u). \tag{1}$$

This factorization assumes conditional independence between $A$ and $X$, given the parameters $\theta$, and is analogous to related approaches on graphs[9,10]. The factorization in Eq. (1) presents various advantages. First, the parameters in $\theta$ can provide interpretable insights about the mechanism driving hyperedge formation, as we show below. In our case, we focus on community structure, hence we model $u$ to represent the community memberships of nodes. Second, it allows for efficient inference of the model parameters $\theta$, as we show in the Methods section. Third, it allows predicting both $A$ and $X$, which is relevant for example in the case of corrupted or missing data.

Having introduced the main structure of the model, we now describe the expressions of the two factors of the joint probability distribution in Eq. (1).

### Modeling structural information

We model the structural information $A$ by assuming that latent communities control the interactions observed. For this, we utilize the Hy-MMSBM probabilistic model[19], which assumes mixed memberships where nodes can belong to multiple communities. This model flexibly captures various community structures (e.g., assortative, core periphery etc.), scales to large hyperedge sizes and allows incorporating covariates flexibly without compromising the efficiency of its computational complexity, as we explain in the Methods section.

Assuming $K$ overlapping communities, $u$ is an $N \times K$ non-negative membership matrix, which describes the community membership for each node $i = 1, ..., N$. A symmetric and non-negative $K \times K$ affinity matrix $w$ controls the density of hyperedges between nodes in different communities. The hypergraph is modeled as a product of Poisson distributions as:

$$P_A(A \mid u, w) = \prod_{e \in \Omega} \text{Pois}\left(A_e; \frac{\lambda_e}{k_e}\right), \tag{2}$$

where

$$\lambda_e = \sum_{i<j: i, j \in e} u_i^T w u_j = \sum_{i<j: i, j \in e} \sum_{k, q=1}^{K} u_{ik} u_{jq} w_{kq}. \tag{3}$$

The term $k_e$ is a normalization constant, which can take on any positive value. In all our experiments we set its value to $k_e = \frac{|e|(|e|-1)}{2}\binom{N-2}{|e|-2}$, with $|e|$ being the size of the hyperedge. Other parametrizations of the likelihood $P_A(A \mid u, w)$ are possible, e.g., using different generative models for hypergraphs with community structure[18,20], but it is not guaranteed that these would yield closed-form expressions and computationally efficient algorithms when incorporating additional attribute information in the probabilistic model. Similarly, in Eq. (2) we assumed conditional independence between hyperedges given the latent variables, a standard assumption in these types of models. Such a condition could in principle be relaxed following the approaches of refs. [27–29]. We do not explore this here.

### Modeling attribute information

We model the covariates $X$ assuming that the community memberships $u$ regulate how these are assigned to nodes. We then assume that a $K \times Z$ matrix $\beta$ with entries $\beta_{kz}$ regulates the contribution of attribute $z$ to the community $k$. This parameter plays a similar role for the matrix $X$ as the matrix $w$ does for the vector $A$. We combine the matrix $\beta$ with the

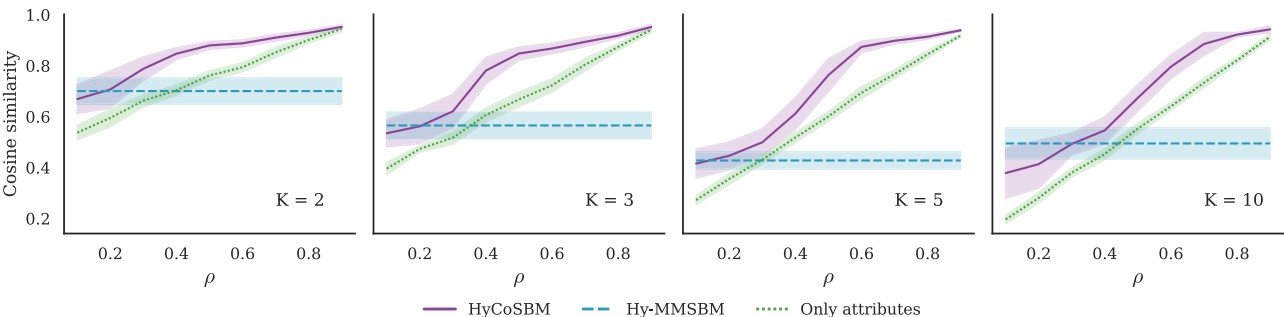

**Fig. 1 | Community detection in synthetic hypegraphs.** We show the cosine similarity between the communities inferred by the various algorithms and the ground truth communities in synthetic hypergraphs, with $N = 500$ and $E = 2720$. We show results for different numbers of communities $K$ (from left to right). The number of attributes $Z$ is selected to be equal to $K$, and the parameter $\gamma$ is set equal to the fraction $\rho$ of unshuffled attributes. We compare HyCoSBM with Hy-MMSBM, which serves as a baseline that only employs structural information. We also measure the cosine similarity of the attribute matrix $X$ and the ground truth membership matrix $u$ Only attributes. Lines and shades around them are averages and standard deviations over 10 different network realizations.

community assignment $u$ via a matrix product that yields the following Bernoulli probabilities:

$$\pi_{iz} = \sum_{k=1}^{K} u_{ik} \beta_{kz}. \qquad (4)$$

We assume that attributes are conditionally independent given the parameters $\pi$, which allows flexibly modeling several discrete attributes at a time. This is implemented by assuming that each entry $X_{iz}$ is extracted from a Bernoulli distribution with parameter $\pi_{iz}$ as:

$$P_X(X|u,\beta) = \prod_{i=1}^{N} \prod_{z=1}^{Z} \pi_{iz}^{X_{iz}} (1 - \pi_{iz})^{(1-X_{iz})}. \qquad (5)$$

To ensure $\pi_{iz} \in [0, 1]$, we constraint $u_{ik} \in [0, 1]$ and $\sum_{k=1}^{K} \beta_{kz} = 1$, $\forall z$.

We focus here on discrete and unordered attributes. This covers many relevant scenarios, including the ones we study in the several real datasets below, e.g., roles of employees in a company or classes of students. Other specific cases could be treated using similar ideas and techniques as the one we propose by suitably modifying the distribution in Eq. (5). We give an example of imposing categorical attributes, when we want to explicitly force that having an attribute of one value does exclude any other possible value, in Supplementary Note C.

### Inference of latent variables

Having defined the probabilistic model Eq. (1) and the two distributions Eqs. (2) and (5), our goal is to now infer the latent variables $u$, $w$ and $\beta$, given the observed hypergraph $A$ and the attributes $X$. To infer these values we consider maximum likelihood estimation and use an efficient expectation-maximization (EM) algorithm that exploits the sparsity of the dataset, as detailed in the Methods section. We combine the log-likelihoods of the two sources of information with a parameter $\gamma$ that tunes their relative contribution, with extreme values $\gamma = 0$ ignoring the attributes and $\gamma = 1$ ignoring the structure, similarly to what has been done in attributed network models[9–11], or in models for information retrieval from text[30,31]. In our experiments, we learn the $\gamma$ hyperparameter from data via cross-validation.

Overall, the inference routine scales favorably with both the system size and the size of the hyperedges, as each EM iteration has a complexity of $O(K(K+Z)(N + |E|))$, which is linear in the number of nodes and hyperedges. We refer to our model as HyCoSBM and make the code available online at github.com/badalyananna/HyCoSBM.

### Detecting communities in synthetic networks

Our first experiments are tests on synthetic networks with known ground-truth community structure and attributes. We generate synthetic hypergraphs using Hy-MMSBM[32] as implemented in the library

HGX[33]. We select parameter settings where inference with Hy-MMSBM is not trivial, to better assess the influence of using attributes, see details in Supplementary Note A. After the networks are created, we generate discrete attributes that match the community membership a fraction $\rho$ of the time, while the remaining fraction $1 - \rho$ are randomly generated. This allows to vary the extent to which attributes correlate with communities and hence the difficulty of inferring the ground truth memberships. We varied $\rho \in [0.1, 0.9]$, with higher values implying that inference of communities is aided by more informative attributes.

As a performance metric, we measure the cosine similarity between the membership vectors recovered by our model and the ground truth ones. In Fig. 1 we can see that, when the attributes are correlated with ground truth communities, HyCoSBM performs better than using either of the two types of information alone. In addition, the performance of HyCoSBM increases monotonically with increasing correlation between attributes and ground truth. Although this is observed also when using attributes alone, the performance of HyCoSBM in recovering the ground truth communities is always higher.

This behavior is consistent across different values of $K$, with larger performance gap between results at low and high $\rho$ at larger $K$, where there are more choices to select from.

In short, these results demonstrate that the model is successfully using both attribute and structural information to improve community detection.

### Results on empirical data

We analyze hypergraphs derived from empirical data drawn from social, political, and biological domains, as detailed in the Methods section. For each hypergraph we describe a different experiment, to illustrate various applications of our method. We select the number of communities $K$ and the hyperparameter $\gamma$ using 5-fold cross-validation. To assess the impact of using attributes, we compare HyCoSBM with three baselines: (i) Hy-MMSBM, that only utilizes the structural information in the hyperedges to detect mixed-membership communities; (ii) HyCoSBM with $\gamma = 0$, which is equivalent to not utilizing the attributes; (iii) HyCoSBM with community assignments $u$ fixed to match the attributes, and only infer the $w$ parameters, which tests how attributes alone perform. Notice that (i) and (ii) differ in that the membership vectors $u$ are unconstrained in Hy-MMSBM, while they are restricted to $u_{ik} \in [0, 1]$ in our model. In (iii) utilizing HyCoSBM and Hy-MMSBM is equivalent, since the two models coincide in the updates for $w$. The results of the following analyses are summarized in Table 1.

Additionally, in the Supplementary Note D we show the advantage of using a hypergraph representation by comparing against results obtained by running a probabilistic model[9] valid on attributed pairwise

**Table 1 | AUC scores on real datasets**

| Dataset | Attribute | $N$ | $|E|$ | $Z$ | HyCoSBM | | | Hy-MMSBM | | Sources |
|---|---|---|---|---|---|---|---|---|---|---|
| | | | | | $K$ | $\gamma$ | AUC | $K$ | AUC | |
| Enron Email | structure | 4423 | 5743 | 2 | 3 | 0.700 | 0.991 ± 0.006 | 2 | 0.913 ± 0.006 | 1 |
| Gene Disease | DPI | 9262 | 3128 | 25 | 30 | 0.500 | 0.9 ± 0.07 | 2 | 0.84 ± 0.122 | 40 |
| High School | class | 327 | 7818 | 9 | 11 | 0.995 | 0.899 ± 0.011 | 24 | 0.884 ± 0.006 | 48 |
| | has filled questionnaire | | | 2 | 21 | 0.800 | 0.892 ± 0.013 | | | |
| | has facebook | | | 2 | 15 | 0.950 | 0.888 ± 0.008 | | | |
| | sex | | | 2 | 16 | 0.800 | 0.889 ± 0.009 | | | |
| Primary School | class | 242 | 12,704 | 11 | 10 | 0.600 | 0.841 ± 0.013 | 11 | 0.841 ± 0.007 | |
| | sex | | | 2 | 12 | 0.200 | 0.841 ± 0.007 | | | 48 |
| Hospital | status | 75 | 1825 | 4 | 2 | 0.200 | 0.776 ± 0.032 | 2 | 0.758 ± 0.016 | 48 |
| Workplace | department | 92 | 788 | 5 | 5 | 0.995 | 0.81 ± 0.02 | 5 | 0.752 ± 0.039 | 48 |
| House Bills | political party | 1494 | 54,933 | 2 | 22 | 0.000 | 0.952 ± 0.003 | 25 | 0.952 ± 0.001 | 49,50 |
| House Committees | political party | 1290 | 335 | 2 | 13 | 0.100 | 0.985 ± 0.015 | 24 | 0.972 ± 0.011 | 51 |
| Senate Bills | political party | 294 | 21,721 | 2 | 23 | 0.000 | 0.929 ± 0.006 | 19 | 0.923 ± 0.003 | 49,50 |
| Senate Committes | political party | 282 | 301 | 2 | 23 | 0.000 | 0.972 ± 0.01 | 21 | 0.963 ± 0.023 | 51 |

We report the AUC scores resulting from 5-fold cross-validation on various real datasets. Values and errors are averages and standard deviations over 5 cross-validation folds. We report the number of nodes $N$, number of hyperedges $|E|$, number of attributes $Z$ and the values of $K$ and $\gamma$ as obtained from cross-validation.

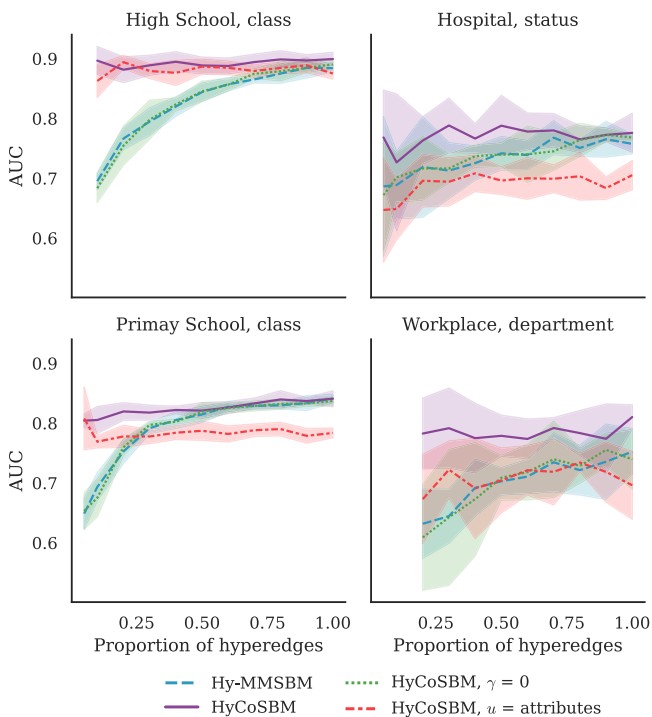

**Fig. 2 | Predicting interactions in close-proximity datasets with partial observations.** We show the performance of various methods in hyperedge prediction tasks, measured by AUC, as we vary the fraction of hyperedges made available to the algorithms. This plot shows that the performance of HyCoSBM remains high when fewer hyperedges are available in input, while that of the algorithms which do not use any attribute drops. Lines and shades around them are averages and standard deviations over 5 cross-validation folds.

networks on a clique expansion, as example dyadic representation of the datasets considered here (we refer to this approach as Clique-Exp). Notice that models valid only on pairwise data do not have a natural expression to measure the probability of a hyperedge of size larger than two. Hence one has to make an arbitrary choice on how to assign this probability from that obtained on pairwise edges. We show results for an example of this choice in the Supplementary Material. HyCoSBM

shows a strong performance in predicting hyperedges, outperforming Clique-Exp in all datasets except two contact datasets of students in schools, where performance is similar. Importantly, Clique-Exp is limited when applied on a biological dataset with large hyperedges, as the corresponding clique expansion contains a much larger number of edges and thus creates a computational bottleneck. Overall, in the datasets considered here, we find no indication that dyadic clique expansions are necessary neither for prediction performance nor for runtime efficiency.

### Recovering interactions on contact dataset

In our first experiment we study human contact interactions, using the data obtained from wearable sensor devices in four settings[8,34–37]: students in a high school (High School) and a primary school (Primary School), co-workers in a workplace (Workplace) and patients and staff in a hospital (Hospital). Hyperedges represent a group of people that were in close proximity at some point in time. Each dataset contains attributes that describe either the classes, the departments, or the roles the nodes belong to.

We measure the ability of our model to explain group interactions by assessing its performance on a hyperedge prediction task. To this end, we infer the parameters using only a fraction of the hyperedges in the dataset. Then we utilize the held out hyperedges to measure the AUC metric, which represents the fraction of times the model predicts an observed interaction as more likely than a non-observed one (higher values mean better performance).

Models that do not utilize any attribute have been previously shown to perform well on such a task on these datasets[18–20] when a large fraction of the dataset was given as input. Here, we vary the amount of structural information available to the algorithms more pronouncedly to assess their robustness in realistic situations where the full data is unavailable and investigate how making use of attributes can compensate for this. To simulate this setting, we delete an increasing fraction of the existing hyperedges (keeping the hypergraph connected) and perform 5-fold cross-validation on the remaining dataset.

The results in Fig. 2 show a significant and monotonic drop in performance for Hy-MMSBM as we decrease the fraction of hyperedges, consequently reducing the amount of structural information available to the algorithm. In contrast, HyCoSBM maintains an almost constant and high performance, all the way down to having access only to 20% of the hyperedges, owing to its usage of the additional attribute

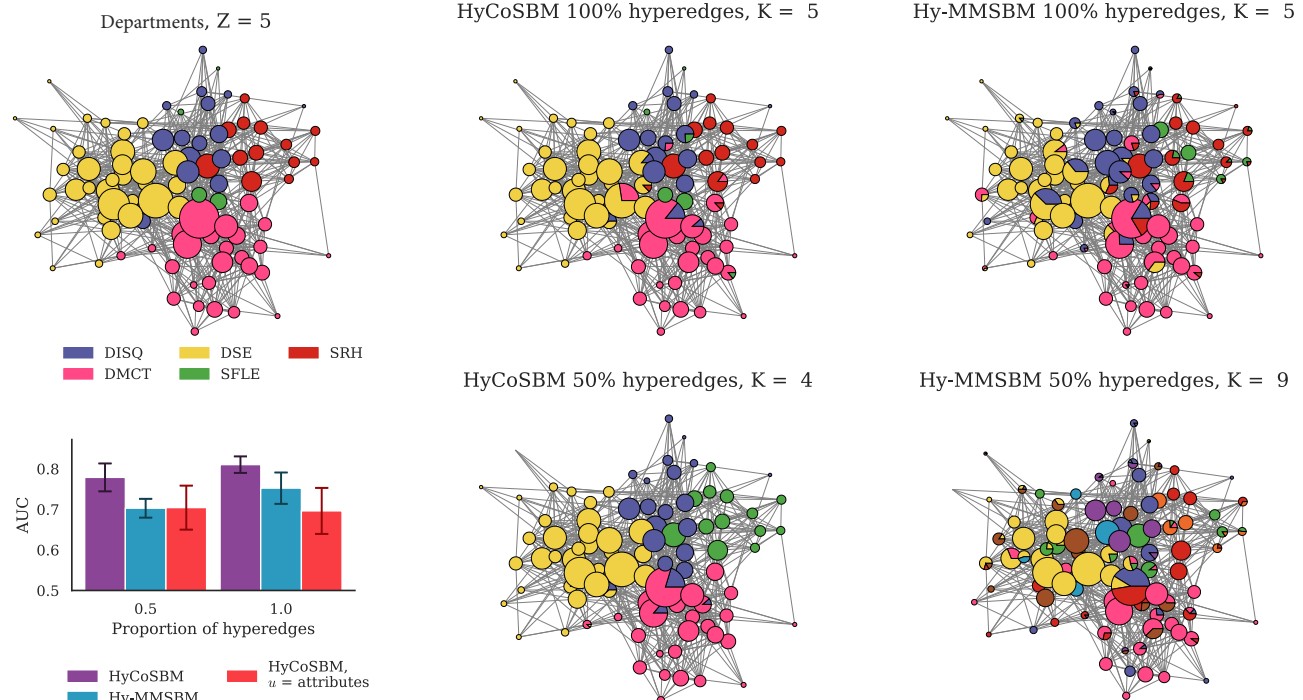

**Fig. 3 | Communities detected in a Workplace dataset from partial observations of close-proximity interactions.** We vary the fraction of hyperedges given in input to the algorithms (top: 100%, bottom: 50%) and compare the inferred communities against the attribute departement (top left). The AUC barplot (bottom-left) shows the performance of the models in hyperedge prediction. Bars and error bars are averages and standard deviations over 5 cross-validation folds. This plot shows that HyCoSBM is able to use the attributes effectively to keep performance high even at a low fraction of input observations.

information. In addition, even in the favorable setting when all hyperedges are available, HyCoSBM yields higher AUC in Workplace (with $\gamma = 0.995$), indicating that incorporating attributes can be beneficial even when robust results are obtained using structural information alone.

Focusing on other datasets where HyCoSBM attains AUC similar to that of other algorithms when all the interactions are utilized, we still observe a difference in the types of communities detected. As an example, in the High School dataset the community assignments $u$ inferred via Hy-MMSBM have cosine similarity of 0.59 with the class attribute of the nodes, as opposed to the cosine similarity of 0.94 observed for HyCoSBM.

These different levels of correlation between inferred communities and attributes, together with observing similar AUC (indicating a similar ability to explain the structural information), could be explained by the presence of competing network divisions, as already observed in network datasets[12,38,39]. Our model allows selecting among divisions, finding ones that correlate with the attribute of interest.

We highlight that, although the communities inferred by HyCoSBM correlate with the attributes, these two are not equivalent. In fact, we observe several cases where the number of detected communities is not equal to the number of attributes. For example, we observe cases where the model detects fewer communities than the number of attributes available. In Fig. 3 the nodes with attribute SFLE (green) are included within the community formed mainly by DISQ nodes (purple) by our model when 50% of the edges are given in input. This partition achieves higher AUC than the model with community assignments fixed and equal to the attributes. In other cases, our model finds smaller communities within the bigger partitions determined by the attributes. We find such an example in the High School dataset in Supplementary Fig. 1, where HyCoSBM finds finer partitions ($K = 11$) than the one given by the $Z = 9$ classes, hierarchically splitting some classes into subgroups. The resulting partition attains a high AUC score. A high number of inferred communities is also observed in Hy-

MMSBM, but, in this case, the AUC drops significantly, and the $K = 30$ communities inferred at 30% of the edges are much more mixed between the classes. In short, the communities inferred by our model do not simply replicate the attribute. Rather, this additional information is used to infer a community structure that better explains the interactions observed in the data.

**Performance with uninformative attributes**

In the previous sections, we have shown how attribute information can aid the recovery of effective communities and improve inference. In general, though, we cannot expect that any type of attribute added to a network dataset may help explaining the observed structure. This may be the case for instance when an attribute is uncorrelated or weakly correlated with the hyperedges, as in the synthetic experiments described above when $\rho$ is close to 0.1.

In this section we study the performance of HyCoSBM in this adversarial regime and show that, when attributes are uninformative, these are readily discarded by our model to only perform inference based on structural information.

To this end, we feed the `sex` and `has facebook` attributes, respectively from the Primary School and High School datasets, into our model. As we show in Fig. 4, the performance of HyCoSBM closely resembles that of the models that do not use any attribute in input, signaling that these attributes are not as informative as `class` to explain the observed group interactions. This is reinforced by a very low AUC for the model that fixes $u$ as the attributes (red line).

We further illustrate this point in four datasets of US representatives. Here, nodes are representatives (in the House of Representatives or in the Senate) and hyperedges represent co-sponsorship of bills (Bills datasets) or co-participation in a committee during a Congress meeting (Committees datasets). The attribute indicates whether the representative is associated with the Republican or Democratic party ($Z = 2$). In Table 2 we show that there is no advantage in using this binary attribute to explain the co-sponsorship nor the co-participation

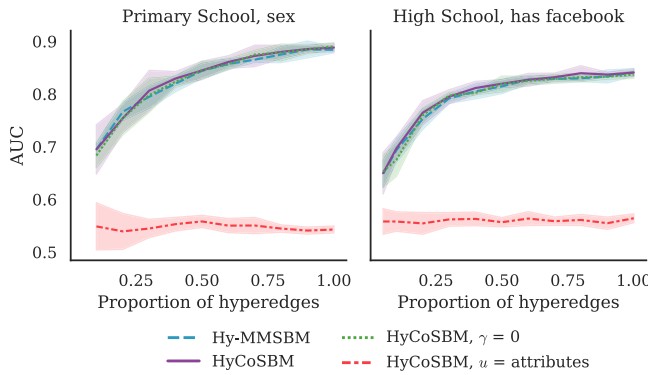

**Fig. 4 | AUC on contacts dataset with partial hyperedges: uncorrelated attributes.** Using sex and has facebook as the attributes, the performance of all models drops as the hyperedges are removed. Lines and shades around them are averages and standard deviations over 5 cross-validation folds.

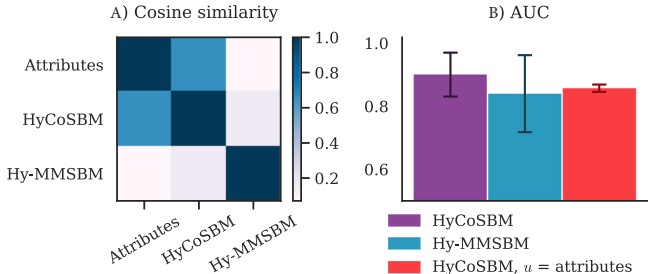

**Fig. 5 | Cosine similarity and AUC in a Gene Disease dataset. A** Cosine similarity between the three types of communities: attribute, HyCoSBM and Hy-MMSBM. **B** AUC in predicting missing hyperedges. Bars and error bars are averages and standard deviations over 5 cross-validation folds. The membership $u$ detected by HyCoSBM correlates with the DPI attribute and achieves higher AUC than both Hy-MMSBM and the model trained with $u$ fixed as the attribute.

**Table 2 | AUC scores on co-sponsorship and co-participation datasets of US representatives**

| Dataset | HyCoSBM | | | Hy-MMSBM | |
|---|---|---|---|---|---|
| | $K$ | $\gamma$ | AUC | $K$ | AUC |
| House Bills | 22 | 0.0 | 0.952 ± 0.003 | 25 | 0.952 ± 0.001 |
| House Committees | 13 | 0.1 | 0.985 ± 0.015 | 24 | 0.972 ± 0.011 |
| Senate Bills | 23 | 0.0 | 0.929 ± 0.006 | 19 | 0.923 ± 0.003 |
| Senate Committees | 23 | 0.0 | 0.972 ± 0.01 | 21 | 0.963 ± 0.023 |

We report the results of cross-validation in terms of selected $K$, $\gamma$, and obtained AUC. Here the node attribute used by HyCoSBM is the political party of the representative (Democrat or Republican, $Z = 2$).

patterns, as the AUC is similar to that of models that do not use attribute information in input. As a confirmation, the value of $\gamma$ obtained via cross-validation is equal to 0 in three out of four cases, and 0.1 in one case, showing that the algorithm tends to discard the attribute information and prefers to rely solely on structural data.

**Improving prediction of Gene-Disease associations**
Our next application is on a biological dataset containing Gene-Disease associations[40]. Here, nodes represent genes, and hyperedges represent a combination of genes specific to a disease. For each node, its Disease Pleiotropy Index (DPI) is available as an attribute, indicating the tendency of a gene to be associated with many types of diseases, with $Z = 25$ possible discrete values. The dataset is highly sparse, as many nodes are present only in one hyperedge. Previous results have shown that inferring missing associations improves sensibly when using all hyperedges in the datasets[19] (with AUC scores up to 0.84), compared to using only hyperedges up to size $D = 25$[18]. In this paragraph, we investigate whether these results can be further improved when additional information is available in the form of the DPI attribute. We find that running HyCoSBM achieves an AUC score of 0.9, indicating that this attribute is informative. Furthermore, we observe that the communities detected by HyCoSBM are similar to those obtained from the attributes, see Fig. 5A, but with a finer division into $K = 30$ communities, which is larger than the $Z = 25$ covariate categories.

**Recovering core-periphery structure with Enron Email dataset**
In this paragraph we focus on the application of our methodology to the Enron Email dataset[41], where nodes represent employees of an organization and hyperedges email exchanges. In particular, the dataset comes with the annotation of nodes being either part of a

"core", which contains employees sending batch emails, or a "periphery", containing the receivers. A hyperedge represents one email batch, and it contains both the core sender and all the periphery receivers. Here, we focus on the study of the core-periphery attribute to predict higher-order interactions in the data.

In this dataset, the core-periphery generative process behind the data is partially known. Hence, it does not come as a surprise that using the "core" and "periphery" labels improves inference and reconstruction. However, our results using HyCoSBM reveal both a more effective and a more nuanced interpretation than that given by using the labels alone. This is because HyCoSBM does not simply replicate the attributes of the nodes, but rather exploits them to achieve an improved inference. To test this hypothesis, we compare three inference scenarios: the vanilla HyCoSBM inference, a constrained version where the attribute matrix $u$ is fixed and equal to the core-periphery assignments, and the Hy-MMSBM algorithm. These achieve AUC scores of 0.99, 0.95, and 0.91, respectively.

As it can be observed in Fig. 6, Hy-MMSBM finds assortative structure dividing the nodes into 2 groups, which is also the number of attributes. Instead, HyCoSBM divides the nodes into three groups: groups 0 and 1, which interact with each other in a disassortative fashion, and group 2, that behaves assortatively. We also observe that the large majority of nodes that have mixed-membership spread in these three groups are core nodes, while periphery nodes have mainly a non-zero membership in group 1. As a result, HyCoSBM unveils a finer-grain division of the core, revealing patterns within it that cannot be inferred by observing the (hard) membership given by the attributes themselves. This is also shown by the inferred $w$ matrix, where core nodes interact mainly with themselves and partially with periphery nodes when we fix $u$ equal to the labels.

In summary, HyCoSBM effectively leverages the data attributes to inform the inference procedure. It does so by exploiting the additional information to extract informative structure and unveiling finer structure than the one given by the observed attributes alone.

**Predicting co-destination patterns in New York City taxi rides**
As a final application, we consider a dataset of taxi rides in New York City[42]. We are interested in measuring patterns of similar destinations, based on travel demands. For this, we consider a given time window and a day of the week and build a hypergraph where nodes are dropoff locations and a hyperedge connects dropoffs that where reached by travelers starting from the same pickup location. Data of this form is often used in urban planning and to understand human mobility co-location patterns[43,44]. The only node attribute available from the data is the "Borough" type (the basic administrative unit in the city of New York), which we utilize in the following experiments. In addition to the

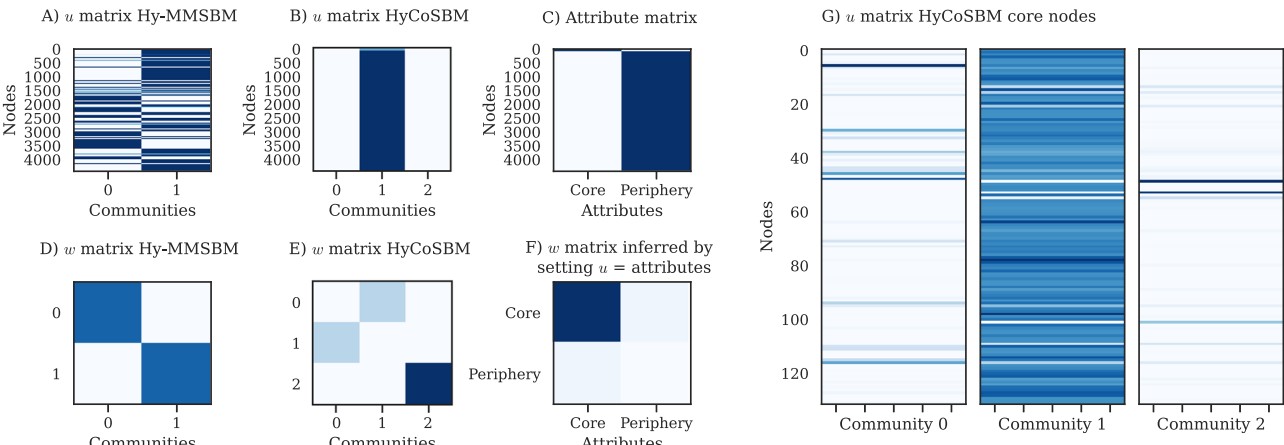

**Fig. 6 | Communities and structure detected in the Enron-Email dataset.**
**A**, **B**, **C** The $u$ matrices inferred by Hy-MMSBM, HyCoSBM and the attributes matrix. The rows $u_i$ for Hy-MMSBM are shown normalized as $u_i / \sum_k u_{ik}$ for better clarity. **D**, **E**, **F** The $w$ matrices inferred by Hy-MMSBM, HyCoSBM and by HyCoSBM when fixing $u$ to the attributes matrix. **G** Zooming in the $u$ matrix for HyCoSBM to highlight the mixed-membership of core nodes. We notice how core nodes have mixed-membership spanning two or three groups. Periphery nodes instead mostly belong only to Community 1 (not shown here).

existing five boroughs, the dataset also contains Newark airport as location. We assign it to a 6th attribute.

We study examples of such a network by considering the week from Saturday November 04 2023 until Tuesday November 14 2023, and building two hypergraphs relative to two different time windows: (i) Monday and Tuesday 06-07.11 between 17.00 and 20.00, and (ii) Saturday and Sunday 04-05.11 between 00.00 and 03.00. These two 3-h time windows are selected to consider diverse travel needs. We expect the first one to capture commuters, the second to capture entertainment and nightlife. We obtain two hypergraphs with $N = 214$ nodes, $E = 523$ and 476 hyperedges, and maximum hyperedge sizes of 132 and 125, respectively, see Table 3.

We assess how informative HyCoSBM is in representing co-destination taxi trips data by comparing it with other approaches in the task of predicting future co-destination locations. Specifically, we train a model on the two datasets described above, and perform hyperedge prediction on analogous datasets built from taxi rides taking place in subsequent days of the same week, in two different time windows. While we expect travel demands to vary with time and day, we also expect correlations to be exploited because of the intrinsic nature of different destination locations, which could attract similar types of passengers in different times and days. To better test this hypothesis, we devise an experiment where for each existing hyperedge $e$ in a given test set (e.g., for the taxi trips of Wednesday and Thursday between 00.00 and 00.03), we extract a non-existing hyperedge $\hat{e}$ where we make a minimal change to $e$. Specifically, we select one node $i \in e$ at random and switch it with another node $j \in V \setminus e$, also selected at random. We refer to this procedure as switch-one-out (SOO). In this way we make the task more difficult as all nodes but one coincide in $e$ and $\hat{e}$. In terms of the Jaccard similarity, we have $J(e, \hat{e}) = |e \cap \hat{e}| / |e \cup \hat{e}| = |e - 1| / |e + 1|$. This construction of the negative test data aims at building challenging comparisons as the prediction of true positives becomes more difficult due to $e$ and $\hat{e}$ being similar. We first run cross-validation on the training datasets and choose the best

parameters. Then we analyze the results using test datasets generated from a different day.

Observing Fig. 7, we find that HyCoSBM achieves a strong performance in predicting future co-destinations consistently across test time frames, which is significantly higher than that of the two comparison methods, Hy-MMSBM and Clique-Exp. The performance gap is higher in predicting co-destinations using the time window of Monday and Tuesday between 17.00 and 20.00 as training set, where the other two approaches have much lower AUC, with Hy-MMSBM attaining higher values than Clique-Exp. We find that HyCoSBM detects communities that are partially aligned with the "Borough" attribute (not shown here). Furthermore, we observe that various node are assigned with mixed-membership spread over more than one community, and that several communities comprise nodes from different boroughs.

## Discussion

We have analyzed how node attributes can be used to guide investigations of higher-order data. We focused on the problem of community detection, introducing a mixed-membership probabilistic generative model for hypergraphs. Our model can explicitly incorporate both hyperedges and node attributes, and find more expressive community partitions by exploiting the combination of these information sources.

We have applied our model to a variety of social, political and biological hypergraphs, showing how prediction of missing interactions can be boosted by the addition of informative attributes, in particular in the regime of incomplete or noisy data. We have also illustrated various scenarios where attributes can be used to select between competing divisions, or cases where they are not informative and can be discarded.

There are a number of possible extensions of this work. One could include additional attribute types, such as attributes on hyperedges, continuous variables or vector variables, for instance considering recent approaches for attributed networks[45]. Similarly, one could consider alternative probabilistic expressions for the structural data, but this would require efforts to derive closed form updates and maintain a low computational complexity. On a related note, our model is based on the assumption that attributes and structure are independent conditionally on the latent variables. This approach is rather general, as the latent variables can potentially take on different semantics. It would be interesting to study other types of dependencies between structure and attributes, as well as investigating in more depth the validity of conditional dependence assumptions in both

## Table 3 | Statistics on hypergraph obtained from NYC taxi drives

| Dataset | $N$ | $|E|$ | $|E_2|$ | $|e|_{max}$ |
|---|---|---|---|---|
| Mon-Tue 17-20 | 214 | 523 | 64 | 132 |
| Sat-Sun 00-03 | 214 | 476 | 53 | 125 |

Number of nodes $N$, number of hyperedges $|E|$, number of dyadic hyperedges $|E_2|$ and maximum hyperedge size $|e|_{max}$.

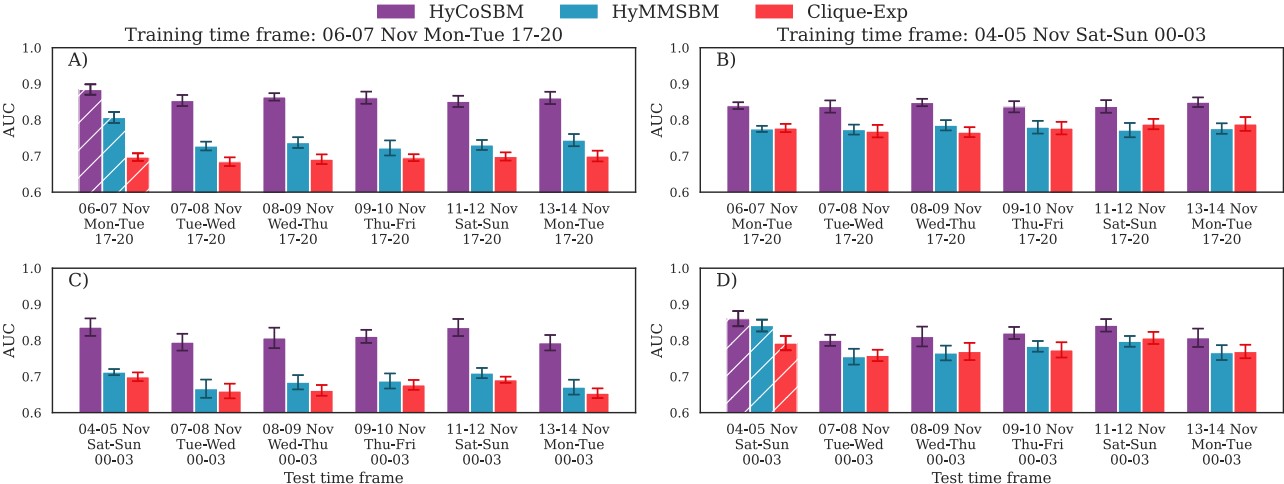

**Fig. 7 | Predicting co-destination taxi rides in New York City.** We report the AUC calculated running SOO hyperedge prediction on hypergraph test sets built considering taxi rides taking place in various time frames subsequent to the two used to train the algorithm. Test time frames with time window between 17.00-20.00

hyperedges and attributes. Finally, our model might be extended to consider dynamical hypergraphs, where communities and interactions can change in time, and assess what role attributes play in this case.

## Methods

### Inference of the latent variables

The likelihood of HyCoSBM factorizes over all hyperedges $e \in \Omega$, and single hyperedges are modeled with a Poisson distribution:

$$P_A(A_e|u,w) = \text{Pois}\left(A_e; \frac{\lambda_e}{k_e}\right). \tag{6}$$

Similarly, the probability of attributes factorizes into Bernoulli probabilities:

$$P_X(X|u,\beta) = \prod_{i=1}^{N} \prod_{z=1}^{Z} \pi_{iz}^{x_{iz}} (1 - \pi_{iz})^{(1-x_{iz})}. \tag{7}$$

Under the Poisson distribution in Eq. (6), it can be shown that the log-likelihood $L_A(u, w)$ of the full hypergraph evaluates to

$$L_A(u,w) = -C \sum_{i<j \in V} u_i^T w u_j + \sum_{e \in E} A_e \log \sum_{i<j \in e} u_i^T w u_j, \tag{8}$$

where $C = \sum_{d=2}^{D} \binom{N-2}{d-2} \frac{1}{\kappa_d}$ and $D$ is the maximum hyperedge size observed[19]. Instead, Eq. (7) yields the log-likelihood

$$L_X(u,\beta) = \sum_{i=1}^{N} \sum_{z=1}^{Z} x_{iz} \log\left(\sum_{k=1}^{K} u_{ik} \beta_{kz}\right) + \sum_{i=1}^{N} \sum_{z=1}^{Z} (1 - x_{iz}) \log\left(\sum_{k=1}^{K} (1 - u_{ik}) \beta_{kz}\right). \tag{9}$$

As we assumed conditional independence of the network part and the attributes part, the total log-likelihood becomes the sum of those two terms. In practice though, performance improves by introducing a balancing parameter $\gamma \in [0, 1]$ that tunes the relative contribution of the two terms[9,11,30,31], yielding a total log-likelihood as:

$$L(u,w,\beta) = (1 - \gamma) L_A(u,w) + \gamma L_X(u,\beta). \tag{10}$$

The value of $\gamma$ is not known a priori, and it can be learned from the data using standard techniques for hyperparameter learning. In our experiments, we utilize cross-validation. The $\gamma$ parameter is necessary to better balance the contribution of the structural and covariate information, as the magnitude of the two different log-likelihood terms can be on different scales, with the risk of biasing the total likelihood maximization towards one of the two terms. This balancing is also useful when attribute data are somehow more (or less) reliable than structural data, for instance when we believe that one is less (or more) subject to noise. Furthermore, $\gamma$ is reminiscent of any hyperparameter of approaches that adjust inference based on prior distributions on the community assignments, as done in some attributed network models, e.g., refs. 12,13.

We note here that the value of $\gamma$ has a clear interpretation only for the extreme cases of 0 or 1, which discards entirely the contribution of one of the two terms. In all the other intermediate cases, its value is not simply interpreted as a percentage contribution of the attributes over the network. This is because $\gamma$ balances the magnitudes of two likelihood terms. In general, the network part is much larger than the attribute one, which draws $\gamma$ to values closer to 1, e.g., 0.995, to compensate for the difference in scales. This does not necessarily mean that the network information is barely used, but rather that it has to be rescaled to allow the attribute information to be effectively considered.

As a final remark, our definition of $X$ allows modeling several discrete attributes at the same time, and the dimension $Z$ is the total number of values, including all the attribute types. Formally, $Z = \sum_{p=1,\ldots,P} z_p$, where $P$ is the number of attribute types (e.g., age and class would give $P = 2$), and $z_p$ is the number of discrete values an attribute of type $p$ can take. Alternatively, the presence of more than one attribute can be modeled by considering separate terms $L_X$, each with a different multiplier $\gamma$. While this formulation would allow for tuning the contribution of attributes more specifically, this comes at a price of higher model complexity (in case of using different expressions for the $L_X$) or higher computational complexity, as one needs to cross-validate more than one type of $\gamma$. We do not explore this here.

**Variational lower bound.** To maximize the total log-likelihood in Eq. (10) we adopt a standard variational approach to lower bound the summation terms inside the logarithm. Introducing the probability distributions $\rho_{ijkl}^{(e)}, h_{izk}$ and $h'_{izk}$ and using Jensen's inequality

$\log \mathbb{E}[x] \geq \mathbb{E}[\log x]$, we get the following lower bounds:

$$\sum_{e \in E} A_e \sum_{i<j \in e} \log \sum_{k,q=1}^{K} \left( u_{ik} u_{jq} w_{kq} \right) \geq \sum_{e \in E} A_e \sum_{i<j \in e} \sum_{k,q=1}^{K} \rho_{ijkq}^{(e)} \log \left( \frac{u_{ik} u_{jq} w_{kq}}{\rho_{ijkq}^{(e)}} \right); \quad (11)$$

$$\sum_{i=1}^{N} \sum_{z=1}^{Z} x_{iz} \log \left( \sum_{k=1}^{K} u_{ik} \beta_{kz} \right) \geq \sum_{i=1}^{N} \sum_{z=1}^{Z} x_{iz} \sum_{k=1}^{K} h_{izk} \log \left( \frac{u_{ik} \beta_{kz}}{h_{izk}} \right); \quad (12)$$

$$\sum_{i=1}^{N} \sum_{z=1}^{Z} (1-x_{iz}) \log \left( \sum_{k=1}^{K} (1-u_{ik}) \beta_{kz} \right) \geq \sum_{i=1}^{N} \sum_{z=1}^{Z} (1-x_{iz}) \sum_{k=1}^{K} h'_{izk} \log \left( \frac{(1-u_{ik})\beta_{kz}}{h'_{izk}} \right); \quad (13)$$

with equality reached when

$$\rho_{ijkq}^{(e)} = \frac{u_{ik} u_{jq} w_{kq}}{\lambda_e}; \quad (14)$$

$$h_{izk} = \frac{\beta_{kz} u_{ik}}{\sum_{k'} \beta_{k'z} u_{ik'}}; \quad (15)$$

$$h'_{izk} = \frac{\beta_{kz}(1-u_{ik})}{\sum_{k'} \beta_{k'z}(1-u_{ik'})}; \quad (16)$$

respectively.

Plugging Eq. (11) into Eq. (8) yields a lower bound $\mathcal{L}_A$ of the structural log-likelihood

$$\mathcal{L}_A(u,w,\rho) = -C \sum_{i<j \in e} u_i^T w u_j$$
$$+ \sum_{e \in E} A_e \sum_{i<j \in e} \sum_{k,q=1}^{K} \rho_{ijkq}^{(e)} \log \left( \frac{u_{ik} u_{jq} w_{kq}}{\rho_{ijkq}^{(e)}} \right). \quad (17)$$

Similarly, Eqs. (12)–(13) yield a lower bound $\mathcal{L}_X$ of the log-likelihood of the attributes:

$$\mathcal{L}_X(u,\beta,h,h') = \sum_{i=1}^{N} \sum_{z=1}^{Z} x_{iz} \sum_{k=1}^{K} h_{izk} \log \left( \frac{u_{ik} \beta_{kz}}{h_{izk}} \right)$$
$$+ \sum_{i=1}^{N} \sum_{z=1}^{Z} (1-x_{iz}) \sum_{k=1}^{K} h'_{izk} \log \left( \frac{(1-u_{ik})\beta_{kz}}{h'_{izk}} \right), \quad (18)$$

so that

$$\mathcal{L} := (1-\gamma)\mathcal{L}_A + \gamma \mathcal{L}_X, \quad (19)$$

is a lower bound of the full log-likelihood.

**Expectation-maximization.** We now aim to optimize the variational lower bound in Eq. (19) with respect to the model parameters $u$, $w$ and $\beta$. To account for the constraint on $\beta$ and $u$, we introduce the Lagrange multipliers $\lambda^{(\beta)}$ and $\lambda^{(u)}$ obtaining the following objective:

$$\mathcal{L}_{constr} := \mathcal{L} - \sum_{z=1}^{Z} \lambda_z^{(\beta)} \left( \sum_{k=1}^{K} \beta_{kz} - 1 \right) - \sum_{i=1}^{N} \sum_{k=1}^{K} \lambda_{ik}^{(u)} u_{ik}. \quad (20)$$

We proceed as in the EM algorithm[46], by alternating two optimization steps until convergence. In one step, we maximize Eq. (20) with respect to the model parameters $u$, $w$, $\beta$ and the Lagrange multipliers $\lambda^{(\beta)}, \lambda^{(u)}$. In the other, we utilize the closed-form updates in Eqs. (14)–(16) for the variational parameters. The procedure is described in detail in Algorithm 1.

Differentiating objective Eq. (20) with respect to the $w$, $\beta$ parameters and the multipliers $\lambda^{(\beta)}$ yields the following closed-form updates:

$$w_{kq} = \frac{\sum_{e \in E} A_e \sum_{i<j \in e} \rho_{ijkq}^{(e)}}{C \sum_{i<j \in V} u_{ik} u_{jq}}, \quad (21)$$

$$\beta_{kz} = \frac{\sum_i (x_{iz} h_{izk} + (1-x_{iz}) h'_{izk})}{\sum_{i,k'} (x_{iz} h_{izk'} + (1-x_{iz}) h'_{izk'})}. \quad (22)$$

Equation (21) is valid when $\gamma \neq 1$ and Eq. (22) is valid when $\gamma \neq 0$.

To obtain the updates for $u$ we distinguish two cases. In the case of $\gamma \neq 0$, differentiating Eq. (20) with respect to $u_{ik}$ yields the condition:

$$a_{ik} u_{ik}^2 - (a_{ik} + b_{ik} + c_{ik}) u_{ik} + b_{ik} = 0, \quad (23)$$

where

$$a_{ik} = (1-\gamma) C \sum_{j \in V, j \neq i} \sum_{q=1}^{K} u_{jq} w_{kq},$$
$$b_{ik} = (1-\gamma) \sum_{e \in E: i \in e} A_e \sum_{j \neq i \in e} \sum_{q=1}^{K} \rho_{ijkq}^{(e)} + \gamma \sum_{z=1}^{Z} x_{iz} h_{izk},$$
$$c_{ik} = \gamma \sum_{z=1}^{Z} (1-x_{iz}) h'_{izk}.$$

The updated values for $u_{ik}$ are found by numerically solving Eq. (23). We take the smallest root of Eq. (23), as this is guaranteed to be in $(0,1)$, as we show in Supplementary Note B. This update automatically yields a value of $u_{ik}$ in $[0,1]$, therefore the constraints on $u$ are inactive and we do not need to differentiate with respect to the Lagrange multipliers $\lambda_{ik}^{(u)}$.

In the case $\gamma = 0$, we differentiate Eq. (20) with respect to both $u_{ik}$ and the Lagrangian multipliers $\lambda_{ik}^{(u)}$ to obtain the update

$$u_{ik} = \frac{\sum_{e \in E: i \in e} A_e \sum_{j \neq i \in e} \sum_{q=1}^{K} \rho_{ijkq}^{(e)}}{C \sum_{j \in V, j \neq i} \sum_{q=1}^{K} u_{jq} w_{kq} + \lambda_{ik}^{(u)}}, \quad (24)$$

which is exactly the same as those of the Hy-MMSBM model[19], except that in our case we have $\lambda_{ik}^{(u)}$ which constrains $u_{ik} \in [0,1]$. Thus, our model is as powerful as Hy-MMSBM when $\gamma = 0$, but, when the attributes correlate well with the communities, our model can utilize this information to boost performance. In practice, in the latter case, cross-validation would yield $\gamma > 0$.

The EM algorithms finds a local maximum for a given starting point, which is not guaranteed to be the global maximum. Therefore, the algorithm is run several times and the best parameters are chosen based on the run that gives the highest log-likelihood.

A pseudocode for the algorithmic implementation is given in Algorithm 1.

**Algorithm 1. HyCoSBM: EM algorithm**
**Inputs:** hypergraph $A$, covariates $X$, hy perparameters $\gamma$ and $K$
**Outputs:** inferred $(u, w, \beta)$
$u, w, \beta \leftarrow init(u, w, \beta)$ : Randomly initialize the parameters
**while** convergence not reached **do**
  $\rho, h, h' \leftarrow update(\rho, h, h')$ ▷ Eqs. (14)–(16)
  $u \leftarrow update(u)$ ▷Eq. (23) or Eq. (24)
  **if** $\gamma \neq 1$ **then**
    $w \leftarrow update(w)$ ▷Eq. (21)
  **end if**
  **if** $\gamma \neq 0$ **then**
    $\beta \leftarrow update(\beta)$ ▷ Eq. (22)
  **end if**
**end while**

## Hyperedge prediction and cross-validation

For all experiments with real datasets we used 5-fold cross-validation with the test AUC as performance metric to select the hyperparameters $K$ and $\gamma$. We varied $K \in \{2, \dots, 30\}$ and $\gamma \in [0.0, 1.0]$. The set of hyperedges was split into 80% and 20% for training and testing. The AUC is calculated by comparing the Poisson probabilities assigned to a given existing hyperedge against that of a randomly generated hyperedge of the same size. Since comparing all possible pairs of observed-unobserved edges is unfeasible, we estimate the AUC via sampling. For every observed edge in the dataset, we draw an edge of the same size uniformly at random, and compute the relative Poisson probabilities. The resulting Poisson probabilities are saved in a vector $R_1$ for the observed edges and $R_0$ for the randomly generated ones. We then compute the AUC as

$$AUC = \frac{\sum (R_1 > R_0) + 0.5 \sum (R_1 = = R_0)}{|R_1|},$$

where $\sum (R_1 > R_0)$ stands for the number of times the Poisson probability of the positive hyperedge was higher than the negative one, $\sum (R_1 = = R_0)$ when they were equal, and the total number $|R_1|$ of comparisons made is equal to the number of hyperedges in the test set.

## Data availability

The data that support the findings of this study are publicly available. The contact datasets at http://www.sociopatterns.org/; the political interactions datasets at https://www.cs.cornell.edu/~arb/data/; the gene-disease dataset at[40]; the Enron dataset at[41]; the New York City taxi data at[42].

## Code availability

The open source codes and executables are available at github.com/badalyananna/HyCoSBM and at[47]. The code uses the `HGX` Python library[33].

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

## Acknowledgements
N.R. acknowledges support from the Max Planck ETH Center for Learning Systems. C.D.B. was supported by the Cyber Valley Research Fund.

## Author contributions
A.B. developed the algorithm and performed the experiments. A.B., N.R., and C.D.B. all conceived the research, analyzed the results and wrote the manuscript.

## Funding

## Competing interests
The authors declare no competing interests.
