## [Peer Review File · Nature Communications]

Structure and inference in hypergraphs with node attributesREVIEWER COMMENTS

Reviewer #1 (Remarks to the Author):

The paper describes an inference method that considers structure and node attributes to better represent the observed hypergraph and detect communities. The proposed method is more accurate than using either type of information alone.

The paper is well-written and organized. The figures are of high quality. The methods are well explained, and the experiments support the author's claims. The results are convincing.

My main concern is with the novelty of the paper. On the one hand, the methods are sound and potentially applicable to many real-world problems. Data availability might be a problem initially, but not in the long run, as we have more and more data available. On the other hand, I am not sure about the novelty of the paper. The paper extends the inference method already proposed by one of the authors of this paper for hypergraphs to include metadata (Ref. 18). Moreover, the incorporation of metadata in similar graph problems has also been done.

So my recommendation to the authors is to emphasize the contribution of the paper, or focus on one or more applications, and show examples where their methods not only perform better but also capture new and perhaps unexpected results. If this is not possible, I would recommend publishing in a more specialized journal.

Reviewer #2 (Remarks to the Author):

The paper deals with an interesting question, namely, whether incorporating information about node attributes into a community detection framework for higher-order interactions

could lead to new insights or allow for more flexibility in terms of inspecting group interactions on some particular node's feature.

The paper appears to be technically correct and the authors have presented results to illustrate how the proposed framework performs in both synthetic and real data. However, some shortcomings make this contribution not worth publishing in Nat Comm in its current version. If a substantial revision that takes the points below into account is done, I am open to reconsidering my recommendation.

- As far as I know, the central research question addressed here is open and I would say, worth exploring. I am, however, not persuaded about its relevance in terms of its innovative character. This is mainly because the methodology is heavily based on previous works, which somehow diminishes the degree of innovation of the proposed mathematical framework but also because the authors have not convincingly shown that one needs to implement the proposed algorithm to learn new features of the system being analyzed. The authors should highlight what is new here methodologically and show that it is not just a clever generalization/integration of previous techniques for community detection and statistical inference, but "a must" when analyzing the systems.

- As said, I am not sure that the authors have convincingly shown the specific value of the proposed framework with respect to alternative ways to explore the same question. The paper shows comparison with and without including information about attributes (whether they are informative or not), but does not present a thorough comparison with simpler scenarios in which higher-order interactions are not taken into account. In other words, I believe that to have a complete description of the pros/cons of the proposed method, it should be shown i) that it is worth representing the system in its higher-order version, ii) what would happen when computing communities (with and without attributes) without taking into account the possible higher-order structure, iii) given i), calculating the community structure alone and with information about attributes included. The paper shows iii), but I am not sure that iii) is needed (i.e., why i)?) and that it is better than ii) (i.e., why is it better than the "low-order" version?)

- Assuming that the above point is satisfactorily addressed and that it is shown that one needs to resort to such a framework that incorporates both the higher-order nature and attributes' information, it remains to be shown that the assumptions are good enough. Specifically, there are some assumptions of statistical/conditional independence that cast some doubts about how the method performs when there are attributes that are conditioned to each other. The authors should discuss this latter issue in more detail and show that it does not have any impact on the outcome of the model.

Dear Reviewers,

thank you for the detailed remarks regarding our Manuscript NCOMMS-23-53831 originally entitled “Hypergraphs with node attributes: structure and inference”.

The feedback has improved our paper and we hope that the revised version of our work can now be considered suitable for publication in *Nature Communications*.

Here we summarize the main changes to the manuscript, more details are given below.

- Added in the main manuscript two experiments to highlight the advantage of using our approach in relevant applications. Specifically, we show how our model i) unveils fine-grained community structure on an email dataset (added Fig. 6); ii) significantly boosts hyperedge prediction in co-destination patterns from taxi rides (added Fig. 7, Supplementary Fig. 1 and Supplementary Note E).
- Added in Supplementary Note D comparisons with a relevant network model applied to a graph projection to highlight advantages of using a hypergraph representation. We also added a discussion about this in the main under “Results on empirical data”.

Report of the Reviewer #1:

The paper describes an inference method that considers structure and node attributes to better represent the observed hypergraph and detect communities. The proposed method is more accurate than using either type of information alone. The paper is well-written and organized. The figures are of high quality. The methods are well explained, and the experiments support the author's claims. The results are convincing.

We thank the reviewer for the positive feedback and for appreciating our work.

My main concern is with the novelty of the paper. On the one hand, the methods are sound and potentially applicable to many real-world problems. Data availability might be a problem initially, but not in the long run, as we have more and more data available. On the other hand, I am not sure about the novelty of the paper. The paper extends the inference method already proposed by one of the authors of this paper for hypergraphs to include metadata (Ref. 18).

Thanks for your comment. We would like to point out that combining a structural part with an attribute one to build a joint model for hyperedges and node attributes is not trivial. We did indeed select the structural part of the likelihood as in ref. 18, but it is not trivial then to combine it with the attribute part in a way this is both principled and computationally efficient. As an example, we notice that in HyCoSBM we need to constrain the elements of matrix u to the $[0, 1]$ range, as opposed to the unconstrained optimization of reference 18. In the paper, we propose two different formulations of the model (one which is the focus of the main text, and an alternative for excluding attributes in Supplementary Note C), but it is not straightforward to start from any probabilistic model for hypergraph (to model the structural part) and then combine with a model for capturing the attributes while keeping it usable in practice.

Moreover, the incorporation of metadata in similar graph problems has also been done. So my recommendation to the authors is to emphasize the contribution of the paper, or focus on one or more applications, and show examples where their methods not only perform better but also capture new and perhaps unexpected results. If this is not possible, I would recommend publishing in a more specialized journal.

We agree that adding node attributes to a standard network has been addressed several times in the past. However, as mentioned in the response to the previous point, we believe the technical implementation of the idea to be non-trivial. We also agree with the reviewer's initial stance that the impact of our model on the community is of high potential. In line with the suggestion given here, and in order to showcase how such a model can have relevant consequences in studying higher-order data with attributes, we have two included additional studies. As we comment in the text, we show how these two applications to real data reveal behaviors and patterns that other model could not capture, due to either not considering higher-order or covariate information.

In addition, we added comparisons with a relevant network model on projections into a (pairwise) graph of the datasets studied in the paper, to also show the differences and the advantages in using a specific hypergraph representation.

Report Reviewer #2:

The paper deals with an interesting question, namely, whether incorporating information about node attributes into a community detection framework for higher-order interactions could lead to new insights or allow for more flexibility in terms of inspecting group interactions on some particular node's feature. The paper appears to be technically correct and the authors have presented results to illustrate how the proposed framework performs in both synthetic and real data.

We thank the reviewer for the positive feedback.

However, some shortcomings make this contribution not worth publishing in Nat Comm in its current version. If a substantial revision that takes the points below into account is done, I am open to reconsidering my recommendation.

Our response:

We thank the reviewer for raising these points. We have now updated the manuscript incorporating your feedback.

As far as I know, the central research question addressed here is open and I would say, worth exploring. I am, however, not persuaded about its relevance in terms of its innovative character. This is mainly because the methodology is heavily based on previous works, which somehow diminishes the degree of innovation of the proposed mathematical framework but also because the authors have not convincingly shown that one needs to implement the proposed algorithm to learn new features of the system being analyzed. The authors should highlight what is new here methodologically and show that it is not just a clever generalization/integration of previous techniques for community detection and statistical inference, but "a must" when analyzing the systems.

Thanks for your comment. We would like to point out that combining a structural part with an attribute one to build a joint model for hyperedges and node attributes is not trivial. We did indeed select the structural part of the likelihood as in ref. 18, but it is not trivial then to combine it with the attribute part in a way this is both principled and computationally efficient. As an example, we notice that in HyCoSBM we need to constrain the elements of matrix u to the $[0, 1]$ range, as opposed to the unconstrained optimization of reference 18. In the paper, we propose two different formulations of the model (one which is the focus of the main text, and an alternative for excluding attributes in Supplementary Note C), but it is not straightforward to start from any probabilistic model for hypergraph (to model the structural part) and then combine with a model for capturing the attributes while keeping it usable in practice.

We followed your suggestion and added two new investigations of real applications that reveal behaviors and patterns that methods that either do not consider the attributes or do not represent the structure as an hypergraph could not capture. In addition, we added comparisons with a relevant network model on projections into a (pairwise) graph of the datasets studied in the paper, to also show the differences and the advantages in using a specific hypergraph representation.

As said, I am not sure that the authors have convincingly shown the specific value of the proposed framework with respect to alternative ways to explore the same question. The paper shows comparison with and without including information about attributes (whether they are informative or not), but does not present a thorough comparison with simpler scenarios in which higher-order interactions are not taken into account. In other words, I believe that to have a complete description of the pros/cons of the proposed method, it should be shown i) that it is worth representing the system in its higher-order version, ii) what would happen when computing communities (with and without attributes) without taking into account the possible higher-order structure, iii) given i), calculating the community structure alone and with information about attributes included. The paper shows iii), but I am not sure that iii) is needed (i.e., why i)?) and that it is better than ii) (i.e., why is it better than the "low-order" version?)

Thank you for allowing us to clarify these important aspects of dealing with higher-order data. As pointed out above, we have now added comparisons with a relevant network model on projections into a (pairwise) graph of the datasets studied in the paper, to also show the differences and the advantages in using a specific hypergraph representation.

Point i) asks whether higher-order representations are more useful than dyadic ones. Referring to previous work (1; 2; 3; 4; 5), it has been shown that in many cases decomposing higher-order information into graph representations incurs a relevant loss of information that results in worse, incorrect or different inference conclusions than when using a hypergraph representation. Furthermore, clique decompositions and similar techniques are often simply not computationally feasible as they result in highly dense, at times fully-connected graphs. In the studies that we include in this revised version, we add to this existing literature and find that on most datasets the link prediction performances substantially deteriorate when higher-order information is not taken into account.

Point ii) is connected to the first, and deals with the differences in the communities inferred on hypergraphs vs on dyadic data. In the new section with experiments on co-destination of taxi data we show how learning communities aided by attributes (the same attribute used by our model) using a graph representation gives significantly weaker results in predicting future co-destination, than learning them using the full hypergraph representation. Hence, in this application, downstream analysis is significantly impacted by how the data is represented.

Point iii) is central to the paper and, as the reviewer suggests, has been thoroughly addressed in our studies. We believe the new real-data applications added in the reviewed version to further solidify the validity of our approach.

Assuming that the above point is satisfactorily addressed and that it is shown that one needs to resort to such a framework that incorporates both the higher-order nature and attributes' information, it remains to be shown that the assumptions are good enough. Specifically, there are some assumptions of statistical/conditional independence that cast some doubts about how the method performs when there are attributes that are conditioned to each other. The authors should discuss this latter issue in more detail and show that it does not have any impact on the outcome of the model.

Thanks for raising this. We show in the Supplementary Note C an alternative formulation for the attribute part that is specifically valid for excluding attributes, i.e. when having one attribute excludes the possibility of taking other possible attribute values. While in some cases the attributes considered in the experiments in the main could be cast as excluding attributes, we did run some tests using both attribute models (the one presented in the main and the one in the SI) but did not see clear advantages for the model specific to excluding attributes.

In general, this discussion of the generative assumptions could be made on most independence conditions, for example those between hyperedges, or even on classical SBM model on dyadic networks, or other similar works that consider independence between attributes or structural aspects in dyadic and higher-order data. We have now added a sentence pointing this out more clearly in the discussion. Overall, we find this discussion a broad topic that falls outside the scope of a work proposing an efficient and practically useful generative model like the one here. As a strong validation that our assumptions are approximately met in practice (and as for most probabilistic models) we find the empirical performances presented here to be a strong point in favor of our formulation.

References

- [1] E. Estrada, J. A. Rodríguez-Velázquez, Subgraph centrality and clustering in complex hyper-networks. *Physica A: Statistical Mechanics and its Applications* **364**, 581–594 (2006).
- [2] A. Eriksson, D. Edler, A. Rojas, M. de Domenico, M. Rosvall, How choosing random-walk model and network representation matters for flow-based community detection in hypergraphs. *Communications Physics* **4**, 1–12 (2021).
- [3] P. S. Chodrow, Configuration models of random hypergraphs. *Journal of Complex Networks* **8**, cnaa018 (2020).
- [4] M. Contisciani, F. Battiston, C. De Bacco, Inference of hyperedges and overlapping communities in hypergraphs. *Nature Communications* **13**, 7229 (2022).
- [5] N. Ruggeri, M. Contisciani, F. Battiston, C. De Bacco, Community detection in large hypergraphs. *Science Advances* **9**, eadg9159 (2023).

REVIEWERS' COMMENTS

Reviewer #1 (Remarks to the Author):

As I mentioned in my first review, the proposed method is more accurate than using either type of information alone. The paper is well-written and organized, the methods are well-explained, the experiments support the author's claims, and the results are convincing. The new experiments complement those in the previous version. However, I still have some concerns about Supplementary Note D:

1) On Table I of the Supplementary Material, in some rows, the $|E_{\text{dyadic}}|$ is smaller than $|E|$. However, the clique projection should always have more pairwise links than the original hypergraph.

2) The authors mention that "In contacts datasets, we observe that the majority of interactions are pairwise (i.e., $|E| \approx |E_{\text{dyadic}}|$), with the bulk of the interactions being of sizes two and three. For this reason, we did not observe a significant difference between Clique-Exp and yCoSBM on the High School and Primary School datasets in predicting hyperedges." However, in the Workplace dataset, this difference seems to be even smaller, and the results are improving. I would suggest that the authors expand this discussion a bit.

In summary, the paper is well-written and organized, the methods are well-explained, the experiments support the author's claims, and the results are convincing. The new experiments complement those in the previous version. Therefore, I recommend a minor revision.

Reviewer #2 (Remarks to the Author):

The authors have made an effort to address the comments raised by the reviewers. I believe that they have succeeded in showing new examples and possible applications of their methodology. I also reiterate that the MS appears to be technically correct and that results are of certain interest, although I failed to see the importance of this contribution beyond a specialized audience. The authors have also commented on this in their response, but honestly, I do not see how much conceptual or phenomenological improvement this work represents in relation to the existing literature on which it capitalizes. If the question is whether the paper is a solid technical contribution that could be of interest to a specialized audience, the answer is yes. On the contrary, if the question is whether this contribution represents a breakthrough in the field or an advancement of broad interest, I am inclined to answer no.

I also understand that the latter is a subjective opinion and the authors should have their say as well. Therefore, I believe that whether to accept or reject the MS is more an editorial than a scientific decision.

Dear Reviewers,

thank you for the positive remarks regarding our Manuscript NCOMMS-23-53831 originally entitled “Hypergraphs with node attributes: structure and inference”.

We have now incorporated the remaining points and we hope that the revised version of our work can now be considered suitable for publication in *Nature Communications*.

Here we summarize the main changes to the manuscript, more details are given below.

- Edited Supplementary Table 1 to make more clear the values of the edges and rephrased the related discussion accordingly, following Reviewer’s 1 suggestion.
- Removed Supplementary Figure 1 to avoid any copyright issue. Given that this was marginal and the qualitative explanation behind it remains valid this is not impacting the core message of the manuscript.

Report of the Reviewer #1:

As I mentioned in my first review, the proposed method is more accurate than using either type of information alone. The paper is well-written and organized, the methods are well-explained, the experiments support the author’s claims, and the results are convincing. The new experiments complement those in the previous version. However, I still have some concerns about Supplementary Note D:

We thank the reviewer for the positive feedback and for appreciating our work.

1) On Table I of the Supplementary Material, in some rows, the $|E_{dyadic}|$ is smaller than $|E|$. However, the clique projection should always have more pairwise links than the original hypergraph.

Sorry for the confusion. We have now added a third column with the number of hyperedges of size two, to distinguish more clearly that number from the number of (pairwise) edges generated in the clique expansion. The numbers displayed there are the *unique* edges, not accounting for weight. In this case, if many hyperedges contain similar sets of nodes, then the number of unique edges in the clique expansion can be smaller than the total number of unique hyperedges. An example is a hypergraph with hyperedges: (A,B,C), (A,B),(A,C), (B,C); the first hyperedge generates 3 edges in the clique expansion, but these are already present in the dataset as the original hypergraph already contains these 3 pairs as hyperedges of size 2. Then, the total number of unique hyperedges is 4 but the clique expansion has 3. We have now clarified this better in the Supplementary Material.

2) The authors mention that “In contacts datasets, we observe that the majority of interactions are pairwise (i.e., $|E| \approx |E_{dyadic}|$), with the bulk of the interactions being of sizes two and three. For this reason, we did not observe a significant difference between Clique-Exp and yCoSBM on the High School and Primary School datasets in predicting hyperedges.” However, in the Workplace dataset, this difference seems to be even smaller, and the results are improving. I would suggest that the authors expand this discussion a bit.

Thanks for pointing this out, we have now rephrased that sentence accordingly expanding more on this.

In summary, the paper is well-written and organized, the methods are well-explained, the experiments support the author's claims, and the results are convincing. The new experiments complement those in the previous version. Therefore, I recommend a minor revision.

Thank you for the positive feedback.

Report Reviewer #2:

The authors have made an effort to address the comments raised by the reviewers. I believe that they have succeeded in showing new examples and possible applications of their methodology. I also reiterate that the MS appears to be technically correct and that results are of certain interest, although I failed to see the importance of this contribution beyond a specialized audience. The authors have also commented on this in their response, but honestly, I do not see how much conceptual or phenomenological improvement this work represents in relation to the existing literature on which it capitalizes. If the question is whether the paper is a solid technical contribution that could be of interest to a specialized audience, the answer is yes. On the contrary, if the question is whether this contribution represents a breakthrough in the field or an advancement of broad interest, I am inclined to answer no.

I also understand that the latter is a subjective opinion and the authors should have their say as well. Therefore, I believe that whether to accept or reject the MS is more an editorial than a scientific decision.

Thank you for the positive feedback.